# Fe65: A Scaffolding Protein of Actin Regulators

**DOI:** 10.3390/cells10071599

**Published:** 2021-06-25

**Authors:** Vanessa Augustin, Stefan Kins

**Affiliations:** Division of Human Biology and Human Genetics, Technical University of Kaiserslautern, 67663 Kaiserslautern, Germany

**Keywords:** Mena, ELMO, Tip60, cortactin, DOCK, Arf6, Rac, Arp2/3, neurite outgrowth, structural synaptic plasticity

## Abstract

The scaffolding protein family Fe65, composed of Fe65, Fe65L1, and Fe65L2, was identified as an interaction partner of the amyloid precursor protein (APP), which plays a key function in Alzheimer’s disease. All three Fe65 family members possess three highly conserved interaction domains, forming complexes with diverse binding partners that can be assigned to different cellular functions, such as transactivation of genes in the nucleus, modulation of calcium homeostasis and lipid metabolism, and regulation of the actin cytoskeleton. In this article, we rule out putative new intracellular signaling mechanisms of the APP-interacting protein Fe65 in the regulation of actin cytoskeleton dynamics in the context of various neuronal functions, such as cell migration, neurite outgrowth, and synaptic plasticity.

## 1. The Fe65 Protein Family

In mammals, the scaffolding protein family Fe65 is composed of Fe65 itself and two Fe65-like proteins, Fe65L1 and Fe65L2. They are all encoded by single genes called *APBB1*, *APBB2*, and *APBB3*, respectively [1,2,3]. *APBB* stands for APP-binding family B and refers to the observation that all three Fe65 proteins bind to the amyloid precursor protein (APP; Box 1) involved in the pathogenesis of Alzheimer’s disease (AD) [1,4,5,6,7,8,9]. Initially, the cDNA of Fe65 was cloned from rat brain and later described as a putative transcriptionally active protein with similarity to retroviral integrases [10,11]. Later, the predominant expression of Fe65 in the brain was confirmed by different studies, whereas, both Fe65L1 and Fe65L2 mRNAs were more widely expressed in non-neuronal tissues [1,2,3,7,12]. Thus far, due to limitations of available antibodies, only Fe65 distribution in brain tissue has been studied in detail [13]. Here, Fe65 shows broad expression throughout the brain, increasing from birth to adulthood.

All members of the Fe65 family have a conserved domain structure, with a WW domain and two C-terminal phosphotyrosine-binding domains (PTB1 and PTB2), whereas human Fe65 and Fe65L1 have a long N-terminal domain with a length of 258 and 290 amino acids, respectively. The corresponding region is missing in Fe65L2. In addition, there are several splice variants of all members of the Fe65 family. At least six different isoforms have been reported for Fe65: Isoform 1, also called p97Fe65, is the longest isoform with 710 amino acids; isoforms 2 and 3 lack two amino acids, E462 and R463, resulting from a deletion of mini-exon 9 [14]; isoforms 3, 5 and 6 have an N-terminal deletion of 240 amino acids and instead carrying a short N-terminal sequences of 6–20 amino acids derived from alternative start positions; isoform 4, also called p60Fe65, is N-terminally truncated by 259 amino acids. Remarkably, splice variants 2 and 3, which lack exon 9, are predominantly expressed in non-neuronal cells [14], while p60Fe65 is expressed in neurons but absent from some brain regions, such as the cerebellum [15]. Interestingly, Fe65 is also a subject of proteolytic cleavage, resulting in the product p65Fe65, which has an increased affinity for APP compared to the full-length p97Fe65 [16,17]. For Fe65L1, at least four isoforms are annotated in the database (UniProt), but to our knowledge only three isoforms have been yet experimentally validated [2,12,18]. For Fe65L2, at least six different isoforms are annotated in the database (UniProt), which show variability in the PTB1 domain, but have not yet been experimentally validated.

Fe65 homologs have been reported for many different vertebrates, including humans, mice, and fish. Furthermore, in non-vertebrates, a homolog protein of Fe65, Feh-1 from Caenorhabditis elegans, was reported that also has a conserved domain structure [19,20]. However, the closest homolog to Feh-1 in Drosophila, showing high homology in the PTB domain, is Numb1. This indicates that the gene of Fe65 with a WW and two PTB domains was lost in the group of arthropods. However, more detailed studies will be required to clarify the phylogenetic development of the Fe65 gene family.

Recent X-ray crystallography and NMR measurements suggest a homotypic dimerization of Fe65 via the PTB2 domain, involving unwinding of a C-terminal α-helix at the end of one PTB2 domain, binding to the PTB2 domain of a second Fe65 molecule [21]. This intermolecular PTB2–PTB2 binding might occur simultaneously with the predicted intramolecular WW–PTB2 interaction, which involves the PTB1–PTB2 boundary [22]. Likely, the dimerization property is unique to Fe65, as the structural essential Aspartate 662 and Arginine 665, forming a salt bridge in the dimerization pocket, are not conserved in Fe65L1 and Fe65L2. However, currently it is unclear to what extent Fe65 dimerization might affect the binding of interaction partners and its physiological function. 

Box 1The Amyloid Precursor Protein and its Physiological Function.The amyloid precursor protein (APP) is a type I transmembrane protein with a large extracellular and a short intracellular domain. It undergoes a complex proteolytic processing by sequential cleavage of different sheddases that cleave extracellular portions of transmembrane proteins, releasing the soluble ectodomains, followed by γ-secretase cleavage of the residual membrane tethered stub. Best investigated sheddases are the α- and β-secretase, releasing the soluble fragment sAPPα or sAPPβ, respectively, with only 13 amino acids difference in length but severe differences in function. While sAPPα has clear neuroprotective properties, these were mostly observed in lower activity or not at all in comparative studies for sAPPβ. Depending on the sheddase cleavage, also the fragments released after intramembranous cleavage by γ-secretase differ dramatically. In the so-called amyloidogenic pathway after sAPPβ was cleaved off, two fragments are generated. The APP intracellular domain that gets released in the cytoplasm and the extracellular released Aβ peptide that forms oligomers and large aggregates that accumulate in form of β-amyloid plaques in Alzheimer’s disease. In contrast, in the non-amyloidogenic pathway after sAPPα generation, a non-toxic instable P3 fragment and AICD get released. Notably, the AICD is proposed to be involved in regulation of transcriptional activity together with Fe65. However, extensive research revealed more complex APP processing, involving a large variety of different secreted factors (>10) with diverse neuroprotective or pathogenic properties. Supplementary to the function of the secreted fragments, additional functions of full-length membrane-bound APP have been proposed as co-receptors for very different signaling pathways, involved in neurite outgrowth and synaptic plasticity. Interestingly, a function of APP forming transcellular dimers as a synaptic adhesion molecule has also been suggested in this context. However, the molecular signaling of APP is not yet understood, genetic studies, particularly considering the overlapping function of the two APP homologous proteins, APLP1 and APLP2, clearly showed an essential contribution in diverse cellular functions such as neuronal outgrowth, synaptic plasticity, and vesicular trafficking [23,24,25,26].

## 2. Fe65 Interaction Partners

Important insights into the putative function of the Fe65 family come from the analysis of interaction partners. Most attention has been paid to the Fe65-PTB2 domain as it directly interacts with the APP C-terminus that links Fe65 to a central protein of AD [6]. Notably, the mode of interaction observed for Fe65 homo-dimerization mimics the interaction of the APP-C-terminus with the Fe65-PTB2 domain, indicating that Fe65 dimerization might prevent low-affinity interactions. The binding to APP might cause a switch to an active monomeric Fe65 state [21].

In addition to APP, more than 20 different Fe65 interaction partners have been reported [27] that can be clustered into different functional groups. A central protein in one of these clusters is the histone acetyltransferase Tip60, possibly forming a transcriptional active ternary complex with the liberated APP intracellular domain (AICD), AICD/Fe65/Tip60, allowing transition of Fe65 from a closed to an open active conformation [28,29]. Interestingly, solely Fe65 in complex with APP is capable of regulating gene expression, while co-expression of APP together with Fe65L1 or Fe65L2 did not mediate transcriptional activity and did not cause an AICD translocation to nuclear spots [28,30,31,32,33]. In line with the assumption that the AICD/Fe65 complex is involved in transcriptional regulation, Fe65-PTB1 binding to a transcription factor, CP2/LSF/LBP1, was reported [34]. Moreover, the Fe65 WW domain was found to interact with Abl tyrosine kinase and the nucleosome assembly factor SET, which plays an intriguing role in nuclear signaling and transcriptional activation [30,35,36]. Different target genes activated by the AICD/Fe65 complex were proposed, including KAI1, APP, and also actin cytoskeleton regulators, such as alpha-actin2 and transgelin [33,37,38,39]. Despite the strength of data clearly pointing to a role of AICD and Fe65 in transcriptional regulation [40,41], the topic is still controversially discussed [41,42,43], mainly because the precise mode of action and the relevant target genes have yet to be defined.

A second group of interaction partners links Fe65 function to lipid metabolism. For instance, the family of low-density lipoprotein receptors (LDLR), including the low-density lipoprotein receptor-related protein 1 (LRP1), very low-density lipoprotein receptor (VLDLR), Megalin/LRP2, and ApoEr2 were shown to bind to the Fe65 PTB1 domain [44,45,46,47,48]. This putative link of Fe65 to ApoE is of particular interest, as ApoE4 has been reported as a major risk factor for AD [49].

A third group of interaction partners was identified in a mass spectrometry-based analysis, reporting more than a hundred Fe65 interaction partners, including some involved in calcium regulation [50]. In this context, it is worth mentioning that Fe65 was reported to bind to P2X2 receptors, involved in synaptic plasticity of excitatory synapses [51]. Together, these data clearly indicate that the adaptor protein Fe65 is functionally involved in distinct cellular processes. However, how these competing interactions of different binding partners get regulated is not clear yet and will require further in vivo analysis to enlighten the interplay of the multiple at least partially competing interaction partners in a physiological context.

## 3. Fe65 Associated Actin Cytoskeleton Regulators

Another major group of Fe65 interacting proteins is involved in actin regulation, encompassing Mena (mammalian ENA), cortactin, and ELMO1/DOCK1 (Table 1) [52,53,54,55,56]. In the following, we will discuss the role of these proteins in cellular functions to unravel putative mechanistic links of Fe65 to the actin cytoskeleton.

### 3.1. Mena

Mena, the mammalian homolog of Drosophila enabled, is a member of the Mena/VASP protein family, consisting of vasodilator-stimulated phosphoprotein (VASP), Ena-VASP-like protein (EVL), and Mena. All members are composed of two Ena/VASP homology domains (EVH1 + EVH2) and a proline-rich core region [59,60,61,62,63].

Mena/VASP members are highly abundant in the brain and concentrated in regions of high actin dynamics such as focal adhesions, stress fibers, lamellipodia, filopodia, ruffles, and growth cones [59,64,65,66,67,68,69,70,71,72,73]. In these processes, they are supposed to have an important physiological function in neurite positioning, outgrowth, axon guidance, and cell movement [71,72,74,75,76,77,78,79,80].

Mena and its homologs act as anti-capping proteins and support filamentous actin polymerization [63,81,82,83]. The association of G-actin to Mena/VASP proteins was found to be tenfold higher in the presence of profilin, and the loading efficiency of profilin with G-actin was increased in the presence of the Mena/VASP family [84,85]. Thus, Mena/VASP proteins are assumed to stabilize and recruit the polymerization competent profilin/G-actin complex to the elongation site of filopodia. Profilin accelerates the exchange of ADP to ATP in G-actin leading to the replenishment of activated ATP-actin pool subsequently favoring actin polymerization [85]. In addition, the Mena/VASP family was shown to reduce the frequency of actin filament branching, mainly provided by the Arp2/3 complex [81,86,87,88]. However, it is unclear whether the reduced actin branching is an active inhibitory process of Mena/VASP or a consequence of competition with monomeric actin. Although, Skruber et al. demonstrated a concentration-dependent profilin manner of actin regulation inducing filopodia formation at low profilin concentrations and additional Arp2/3-dependent lamellipodia formation at higher profilin concentrations [89]. Furthermore, Mena protein family activity can be modulated by posttranslational modifications like phosphorylation and ubiquitination [67,78,90]. Thus, fast and refined remodeling and adaptation during neuronal development can be achieved.

Initial evidence that Fe65 may be involved in actin cytoskeleton regulation comes from a biochemical study showing that Mena preferentially interacts with the WW domain of Fe65 via two central PPxPP motifs, analyzed in detail by crystallography [20,57]. Consequently, the connection of Fe65 with the Mena/VASP family may allow to couple external stimuli to changes in actin cytoskeleton dynamics, similarly as shown for vinculin, zyxin, Robo and semaphorin 6A-1 [64,77,91,92,93], which link the Mena function directly to signals of the extracellular matrix. Although not shown directly, APP interacting with Slit, a repulsive cue and ligand of Robo that promotes filopodia formation while leading to the collapse of lamellipodia structures for the right pathfinding, might involve Fe65 scaffolding activity [55,94]. Thus, Fe65 binding to cell surface proteins, such as APP, ApoEr2 or LDL receptors, may recruit Mena to the plasma membrane, promoting actin polymerization. Consistently, P. Greengard’s lab reported binding and co-localization of Mena with Fe65, APP, and β1-integrin in mobile lamellipodia and focal complexes [52].

### 3.2. Cortactin

Cortactin, a class II nucleation promoting factor, recruits N-WASP (neural Wiskott–Aldrich syndrome protein), which in turn activates the Arp 2/3 complex by changing the position of Arp2 forming an Arp2–Arp3 short-pitch dimer and generating a new daughter actin filament [95,96]. After initiation, N-WASP dissociates and cortactin stays to further stabilize newly formed filament branching points [97,98,99,100,101]. Cortactin, like Mena, is highly abundant at the leading edge of cells, regulating cytoskeletal remodeling [56,97,102,103,104,105,106,107,108,109,110]. While Mena increases the filamentous polymerization of actin, cortactin promotes the formation of meshwork structures by activating the Arp2/3 complex.

Cortactin undergoes different posttranslational modifications, regulating its activity. Phosphorylation by distinct kinases, including Src, Abl, Arg, Erk, PAK, and PKD [102,105,107,111,112,113,114,115], was shown to promote interaction with actin-binding proteins (ABPs), such as N-WASP and Arp2/3 complex [116], stimulating cortactin activity. Acetylation, in contrast to phosphorylation, lowers the association of cortactin with actin and inhibits its Arp2/3-dependent polymerization function [102,103,116,117,118,119,120].

As Fe65 binds to the histone acetylase Tip60 that is capable of increasing cortactin acetylation, it inhibits the association of cortactin with actin [56,116]. In line with this, it was confirmed by using an inducible knockdown of Fe65 in HEK 293T cells that expression of Tip60 increases acetylation of cortactin in the presence but not the absence of Fe65 [56]. As cortactin was shown to have an important role in dendritic spine plasticity [121,122,123], it is tempting to speculate that inhibition of cortactin activity by Fe65/Tip60 mediated acetylation is involved in structural synaptic plasticity and possibly also in cell migration.

In addition to the regulation of actin dynamics, the interaction of cortactin and Fe65 might also play a role in the nucleus, as acetylated cortactin gets translocated to the nucleus, similarly as shown for the Fe65-APP-Tip60 tripartite complex [28,30,32,33,124]. However, the nuclear function of cortactin and its interplay with Fe65 is still unclear.

### 3.3. ELMO1/DOCK1/Arf6/Rac

Rac (Ras-related C3 botulinum toxin substrate 1) is a member of the Rho family of small GTPases that regulates axon and dendrite differentiation, prolongation, and arborization while it is antagonized by RhoA activity [125,126,127]. Rac proteins, like any other small GTPases, cycle between GDP- and GTP-bound (inactive/active) states. Active GTP-bound Rac proteins activate the Arp2/3 complex via the WASP, N-WASP, and WAVE proteins that, in turn, promotes actin polymerization.

Rac proteins get activated by guanine exchange factors (GEFs) that promote nucleotide exchange from GDP to GTP [128]. The GEFs for Rho/Rac GTPases are divided into the Dbl and DOCK (dictator of cytokinesis) families. Dbl family members can activate all members of the Rho family, whereas DOCK GEFs specifically activate Rac and/or Cdc42. In total, more than 80 GEFs are known, of which only 11 belong to the DOCK family that is subdivided into four groups (DOCK A to D), based on sequence similarity and domain organization. DOCK1, also called DOCK180, belongs to the DOCK A family that activates only Rac1. It has an N-terminal SH3 domain that binds the engulfment and cell motility proteins ELMO1–3 and activates RhoG at the plasma membrane. As DOCK1 binds to PI(3,4,5)P3, this might help to localize the complex to the membrane. However, more recent data suggest that the PIP3 binding site binds preferentially to phosphatidic acid (PA). In response to growth factors, PA is partly generated through hydrolysis of phosphatidylcholine by phospholipase D involved in signaling [129]. The DOCK1/ELMO complex mediates the activation of Rac at the leading edge or focal adhesion sites to form lamellipodia, which further promote cell spreading and migration [130]. Notably, the direction of migration further relies on microtubule stabilization, mediated by ACF7, a partner of ELMO [131].

ELMO proteins were first discovered in a genetic screen to identify components required for engulfment of dead cells and cell motility in C. elegans [132]. ELMO and DOCK often form a complex, which exists in an active and inactive state. Activation of cell surface receptors typically stimulates the GEF activity of ELMO/DOCK complexes, which in turn activate Rac proteins for cell migration. However, the mechanism regulating the ELMO/DOCK complex activity has not been fully understood. It can associate with ARNO/Arf family GTPases at the plasma membrane [133,134] where Arf6 controls endocytosis, actin dynamics, and lipid modifications [135,136]. Interestingly, ELMO/DOCK1 and Rac are also involved in the initiation and maintenance of dendritic spines. Along this line, a knockdown of either ELMO or DOCK1 reduces the formation of spines [137] and overexpression of constitutive active Rac1 induces the transition from filopodia to spine formation, increasing the spine density and α-amino-3-hydroxy-5-methyl-4-isoxazolepropionic acid receptor (AMPAR) clustering but simultaneously reduces spine head size. In line with this, the dominant negative form of the Rac1 downstream partner PAK leads to a decrease in spine density and an enlargement of synapses [138]. PAK further phosphorylates the LIM-kinase (LIMK) that deactivates cofilin, an important actin depolymerization factor and thereby promotes profilin-actin polymerization pathways [139,140,141,142,143,144]. Additionally, PAK can also activate Arp2/3 complex-dependent actin meshwork formation.

The interaction of Fe65 and ELMO1 was identified by the group of Dr. Lau [54,58,145,146], showing the binding of ELMO1 to the most N-terminal domain of Fe65, not present in Fe65L2. Interestingly, Fe65 binding releases the intramolecular autoinhibition of ELMO1, which in turn recruits DOCK1 [54,58,146,147]. Furthermore, Fe65 was shown to form presumably a quadripartite complex with Arf6 and ELMO1/DOCK1 facilitating retargeting of ELMO1 to the plasma membrane via involvement of Arf6 [58]. The ELMO1/Fe65/DOCK1/Arf6 complex subsequently activates Rac by GTP loading. 

## 4. Scaffolding Protein as Actin Regulators

The basic idea of scaffolding protein functions is that they bring components of a signaling cascade together into spatial proximity. This can increase the efficiency of signal transduction as well as signal specificity. This function appears crucial as modeling has shown that kinases in a cascade without scaffold proteins have a higher probability of being dephosphorylated by phosphatases before they are even able to phosphorylate downstream targets [148]. In this scenario, scaffold proteins protect active signaling molecules from inactivation or in similar constellations from degradation. They may also act as molecular switches, as interaction with signaling proteins can cause allosteric changes, resulting in signaling activation/inactivation [149].

One well-studied example of a scaffolding protein involved in actin cytoskeleton regulation is β-catenin, which binds to the cytoplasmic end of E-cadherin and to α-catenin, which interacts with the underlying actin cytoskeleton [150]. Cadherin-mediated cell–cell adhesion is thought to also couple the subcellular actin cytoskeleton mechanically between two cells in this way. Such a function implies a stable complex between E-cadherins and the actin cytoskeleton. Remarkably, it has been shown that monomeric α-catenin cannot bind to F-actin and β-catenin simultaneously [151]. Moreover, the monomeric α-catenin preferentially binds β-catenin, whereas the dimeric form competes with Arp2/3 for binding to F-actin, suppressing Arp2/3 activity and favoring actin fibril bundling [152]. In line with this, FRAP analyses at epithelial cell junctions showed a threefold higher dynamic for actin than for E-cadherin, β- and α-catenin, showing similar dynamics [151,152]. Together, these data indicate a dynamic rather than a stable link between the adhesion complex and the actin cytoskeleton [153].

Although not yet shown experimentally, it appears well feasible that Fe65 also functions as a highly dynamic adaptor protein. First, Fe65 was shown to form dimers via the PTB2 domain using the interaction site of the APP-C-terminus [21], indicating that Fe65 might switch between a monomeric membrane-bound and dimeric-free cytosolic state with different interaction partners and functions. Second, the diverse set of above discussed Fe65 interaction partners, might not bind simultaneously, but instead the complex formation is most likely regulated in a dynamic, spatial, and temporal manner by intra- and extracellular signaling events.

Since Fe65 is a phosphoprotein, it seems quite conceivable that phosphorylation is involved in regulating spatial and temporal Fe65 dynamics. Fe65 can be phosphorylated by various protein kinases, including extracellular signal-regulated kinase (ERK1/2), serum- and glucocorticoid-regulated kinase (SGK1), Abelson tyrosine kinase (c-Abl), ataxia telangiectasia mutated/ataxia-telangiectasia- and Rad3-related protein (ATM/ATR) kinase, and glycogen synthase kinase 3β (GSK3β) [36,154,155,156,157]. Interestingly, Fe65 phosphorylation by GSK3β or SGK1 was shown to affect APP processing [155,156], possibly caused by impacts on Fe65 homotypic dimerization [156]. Moreover, phosphorylation of Fe65 at S228, T547, and S566 by ATM/ATR, c-Abl, and SGK1, respectively, affects its nuclear activity [36,154,158,159]. Most interestingly, kinases like ERK1/2 and c-Abl were also shown to be important regulators for actin polymerization [74,160,161], supporting the hypothesis that at least a part of the impact of those kinases on actin cytoskeleton depends on altered regulation of the Fe65 function. However, different kinases also target Fe65 binding partners, such as APP or LRP, which in turn influence the binding affinity to Fe65 [162,163,164,165]. The current understanding of these processes is very much in the beginning and the exact regulatory mechanisms, in particular in highly dynamic processes, such as actin dynamics, need further detailed investigations.

## 5. Genetic Evidence for Fe65 Function in Actin Cytoskeleton Regulation

Important insights into the Fe65 function were gained by analyses of genetically modified mice [15,166,167,168,169]. In addition to the Fe65 family KO mice, mice overexpressing Fe65 together with APP or APP fragments, such as AICD, were also analyzed [170,171]. Those studies showed an impact of Fe65 on APP processing, not observed in Fe65 KO mice, and highlight a function of Fe65 and AICD in neuronal survival and synaptic plasticity, possibly caused by upregulation of GSK3β activity that in turn affects actin polymerization. However, based on these combined overexpression studies, it is challenging to assign specific functions to either Fe65 or APP. To decipher the phenotypes more clearly, genetic studies of Fe65 transgenic animals with APP KO mice might be beneficial. Therefore, here we like to focus on loss of function studies of APP, Fe65, and interacting actin regulators of the Mena/VASP family. Interestingly, some key features, observed in Fe65 KO mice were also found in mice lacking the APP or Mena/VASP family (Table 2). Thus, Fe65/Fe65L1 DKO as well as APP and Mena/VASP TKO mice all exhibit abnormal ectopic accumulations of neuroblasts, migrating through the basal lamina and pial membrane during brain development [76,167,172], resembling a cobblestone or type II lissencephaly [173,174,175]. Additionally, they all represent failures in axon tract formation and reduction or displacement of Cajal Retzius (CR) cells, resulting in disruption of cortical/meningeal layering. Mena/VASP TKO mice exhibit exencephaly that is also found in two out of 31 APP TKO mice [76,172]. The cause for cortical malformation in APP and Fe65 KO mice is not yet understood but could be well explained by defects in actin cytoskeleton regulation, possibly causing problems in glial endfoot formation, lamination, neuronal migration, or defective recognition of stop signals [75,76,167,172]. Fe65 and its interacting ABPs were also shown to positively influence dendritic and axonal outgrowth by elevation of actin polymerization [54,58,72,75,76,145,167,176,177,178]. Misregulation of these pathways arise in neuronal brain malformations, like impaired decussation of the corpus callosum and hippocampal fiber structures, as observed for Fe65/Fe65L1 DKO, APP, and Mena/VASP TKO mice (Table 2).

Additionally, Fe65 and APP family KO mice show severe learning and memory deficits, resulting likely from impairments of synaptic plasticity (Table 2). Some of these phenotypes might be due to alterations in actin cytoskeleton regulation. However, so far there is very limited information on behavior defects of Mena, ELMO/DOCK1, and cortactin KO mice, making it difficult to draw clear conclusions. Notably, Fe65 and APP family KO mice were reported to exhibit deficits at the neuromuscular junction (NMJ) followed by muscle weakness. Changes in NMJ formation in Mena TKO mice were not yet investigated, but studies of Drosophila NMJs revealed a pre- and postsynaptic abundance and function of Ena [179,180,181]. However, to gain further insights, future genetic studies will be required.

Fe65 and APP can regulate cell motility. Therefore, overexpression of APP or Fe65 in MDCK cells increased the cell migration velocity in a wound-healing assay and co-expression of APP and Fe65 further accelerates cell movement [52]. This suggests that Fe65 might function as a downstream signaling factor of APP in this process. In contrast, Fe65 was demonstrated to inhibit cell motility in MDA-MB-231 breast cancer cells [56]. The reason for these contradictious results could be due to differences of the used cell lines and/or target protein expression levels. Overexpression of Ena/VASP proteins in fibroblasts also decreases cell motility in a dose-dependent manner and in line with this, deletion of Ena/VASP family increased fibroblast motility [71]. However, in contrast to these studies, experimental data from Listeria monocytes, Drosophila hemocytes, mouse fibroblasts, B16-F1 mouse melanoma cells, and MTLn3 cells demonstrated a positive regulation of actin-dependent cell movement by the Ena/VASP family [182,183,184,185,186]. Although the impact of Fe65, APP, and Mena on cell motility is quite obvious, the results from the different cell lines cannot be easily compared.

In line with changes of the motility and of the plasma membrane traction, Fe65 was also shown to suppress invasion by binding to Tip60 and cortactin [56]. Actin-dependent invasive capabilities are often related to cancer. Hence, it is not surprising that Fe65, APP, Mena, cortactin, and ELMO/DOCK180/Rac are involved in suppressing or supporting different kinds of cancerous signaling cascades, for example, in breast, thyroid, colon, lung, and pancreas cancer [56,128,185,187,188,189,190,191,192,193,194,195]. It was astonishing that especially breast cancer was highly investigated in association with Fe65 interaction partners (APP, Mena, cortactin, ELMO/DOCK180/Rac) separately but not in a common pathway [56,187,188,189,190,191,196,197,198,199,200,201,202,203,204].

To unravel the functional interdependence of Fe65 and its interaction partners in cell motility, additional genetic studies using the same cell lines and comparable mouse models will be necessary.

## 6. Future Perspectives of Fe65 in Actin Dynamics

Overall, there is clear evidence for a central function of the APP-binding protein family Fe65 in actin regulation. Fe65 likely supports the formation of branched and unbranched actin polymerization, affecting multiple cellular functions, including axonal/dendritic outgrowth, cell migration, structural synaptic plasticity, and intracellular vesicular trafficking. Those diverse functions could be mainly explained by Fe65 interaction with central regulators of actin dynamics. First, Fe65 binds to Mena, which promotes actin elongation by enhancing profilin function. Increased actin polymerization then favors the growth of unbranched actin filaments, leading to filopodia formation, important for environment scanning and positioning as well as for dendritic spine initiation. However, depending on the availability of Arp2/3 complexes, Mena/profilin may also contribute to increasing actin meshwork formation (Figure 1). Furthermore, Fe65 can form complexes with ELMO1, DOCK1, and Arf6 that favor plasma membrane targeting and activation of Rac1. This small Rho GTPase leads to the inhibition of actin severing by phosphorylation of cofilin and the induction of Arp2/3-dependent branched actin polymerization, promoting the formation of lamellipodia and the maintenance and maturation of dendritic spines (Figure 1). Notably, Fe65 interacts with cortactin and might facilitate its acetylation by recruitment of Tip60, which negatively influence Arp2/3-dependent actin formation. However, to fully understand the relationship between cell shape and cell function, we must better understand the transitions between different types of actin networks, important for many cellular processes like cell outgrowth and locomotion, spine plasticity, and intracellular vesicular motility. Here, it will be fundamental to unravel the dynamics and the regulation of the distinct Fe65 complexes.

In addition to the question of how the scaffolding protein Fe65 helps to orchestrate the reorganization of the actin cytoskeleton, it will be central to understand what the upstream regulators of Fe65 are. Based on the high-affinity binding of the Fe65 family members to the APP family, and due to widely overlapping phenotypes in APP/APLP1/APLP2 TKO and Fe65/Fe65L1 DKO mice, it appears very likely that APP/APLPs are important upstream regulators of Fe65 family members (Table 2) [1,4,5,6,7,8,9,23,234]. Consistently, Fe65 and APP are highly abundant in growth cones, are present at the pre- and postsynapse, and were both shown to affect synaptic plasticity [52,53,235,236,237,238,239,240,241,242,243]. The different molecular signaling pathways clearly require more than the formation of one stable complex, but how the functions of Fe65, APP, and their family members are regulated is still not understood. This is further complicated by the fact that Fe65 can functionally link APP with VLDL, ApoEr2, and LRP modulating both the ApoE receptor and APP trafficking as well as processing [45]. Furthermore, APP was described to interact and to increase NMDA receptor surface localization [242,244,245,246,247] and ApoE receptors were shown to regulate NMDA receptor activity that involves alterations in APP endocytosis and interactions with Fe65 [44,47,48,248,249,250,251,252]. Thus, APP-bound Fe65 might link the NMDA and ApoE receptor activity to the actin cytoskeleton, which in turn translates the extracellular signals to alterations in cell shape and function.

The pathological role of the Fe65 family in AD is extensively investigated, but so far, only minor changes in APP processing and amyloid pathology were reported [14,15,16,47,165,167,170,253,254,255,256,257]. However, the link of Fe65 to ApoE, which is a major risk factor of AD, is very obvious [49]. Thus, a better understanding of the molecular signaling of ApoE receptors to Fe65 might unravel novel potential therapeutic strategies. Another central pathological feature of AD is the early loss of synapses [258,259,260,261,262]. As actin dynamics play a key role in this process, Fe65 might also be involved in AD-associated changes in spine regulation by affecting different pathways [263,264,265,266,267,268,269,270,271,272,273,274].

In summary, we found convincing evidence for an important role of Fe65 in the regulation of the actin cytoskeleton in neuronal development and synaptic plasticity. To better understand the pathophysiological role of the Fe65 family in the different actin-dependent processes and the transitions between the different types of actin networks, it will be important to study the control and dynamics of the different Fe65 complexes in more detail.

## Figures and Tables

**Figure 1 cells-10-01599-f001:**
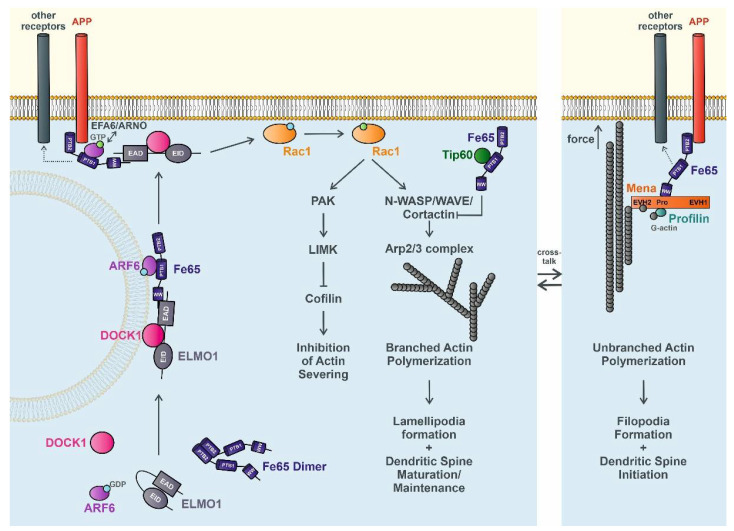
Putative new role of Fe65 in regulating actin dynamics. Fe65 associates with ELMO1/DOCK1 and Arf6 to form a functional complex that is translocated to the plasma membrane and trapped by APP or other potential receptors, such as the ApoE receptor. Arf6-bound GDP gets replaced with GTP via EFA6/ARNO. Subsequent activation of Rac1 induces a cascade inhibiting the severing activity of cofilin and promoting the polymerization of branched actin, which results in the formation of lamellipodia and the induction of dendritic spine plasticity. The Fe65–Tip60 complex may additionally adjust the association of cortactin to actin by acetylation. In a second potential pathway membrane, receptor-bound Fe65 bind to the polymerization-competent complex of Mena, profilin, and monomeric G-actin that supports the elongation of unbranched actin, leading to filopodia formation and dendritic spine initiation. During migration and outgrowth, it is very likely that these processes are regulated dynamically in a homeostasis. ELMO1, engulfment and cell motility protein 1; EAD, ELMO autoregulatory domain; EID, ELMO inhibitory domain; Arf6, ADP-ribosylation factor 6; GDP, guanosine diphosphate; GTP, guanosine triphosphate; WW, tryptophan-tryptophan domain; PTB1/2, phosphotyrosine binding domain 1/2; DOCK1, dictator of cytokinesis 1; APP, amyloid precursor protein; EFA6, exchange factor for Arf6; ARNO, ADP-ribosylation factor nucleotide-binding site opener; Rac1, ras-related C3 botulinum toxin substrate 1; PAK, p21-activated kinase; LIMK, LIM kinase; Tip60, Tat-interacting protein 60 kDa; N-WASP, neural Wiskott–Aldrich syndrome protein; WAVE, WASP family verprolin homologous protein; Arp2/3, actin related protein 2/3; Mena, mammalian enabled; EVH1/2, Ena/VASP homology domain 1/2; Pro, proline-rich region; G-actin, globular actin.

**Table 1 cells-10-01599-t001:** Fe65 interaction partners involved in actin dynamics.

Fe65 Interaction Partner	Fe65-Binding Domain	Putative Function in Actin Dynamics
Mammalian enabled (Mena)	WW [52,57]	Unbranched actin polymerization
Engulfment and cell motility protein (ELMO)	N-Terminus (1–60 amino acids) [54]	Rac1-dependent branched actin polymerization
Tat-interacting protein 60 kDa (Tip60)	PTB1 [56]	Regulation of Cortactin dynamics; inhibition of Arp2/3-dependent actin polymerization
ADP-ribosylation factor 6 (Arf6)	PTB1 [58]	Rac1-dependent branched actin polymerization
Transmembrane receptors, like amyloid precursor protein (APP) and Apolipoprotein E receptor 2 (ApoEr2)	e.g., PTB2 (APP) [1,4,5,6], PTB1 (ApoEr2) [47]	Recruitment of Fe65 to locations of actin dynamics

**Table 2 cells-10-01599-t002:** Systematic summary of phenotypes in Fe65, APP, and Mena/VASP family KOs.

Genotype	Viability		Neuronal Migration/Positioning		Outgrowth		Laminin Organization		Synapse Formation
**p97Fe65^–/–^** (↑p60Fe65 [15])	normal [15]		n. d.		n. d.		n. d.		n. d.
**Fe65^–/–^**	normal [167]		n. d.		n. d.		n. d.		←spine density [169]
**Fe65L1^–/–^**	normal [167]		n. d.		n. d.		n. d.		←spine density [169]
**Fe65^–/–^/Fe65L1^–/–^**	lower Mendelian frequency [167]		lissencephaly type II [167] mislocalization of CR neurons [167] ↓CSPGs in marginal zone heterotopias [167]		↓axonal fibers [167]		↓in marginal zone heterotopias [167] Altered in MEF cells [167] ↑in lens [168]		←spine density [169]
**APP^–/–^** (↑APLP1 [205], ↑APLP2 [205])	normal [206,207,208,209]		impaired axon targeting in retinal ganglion cells [210] impaired axon pruning after whisker plucking in primary somatosensory cortex [211]↑cellular adhesion [212]		↓axonal fibers [207,209,213] ↓axonal/dendritic outgrowth and branching [212,214,215,216,217,218]		n. d.		↓synapse/spine density [212,214,216,217,219,220,221] altered spine plasticity [219,220,221,222,223]
**APP^–/–^ APLP1^–/–^** **APLP2^–/–^**	perinatally lethal [172]		lissencephaly type II [172] rare phenotypes: polymicrogyry, exencephaly [172] disrupted organization and ↓of CR cells [172] disrupted CSPG pattern in marginal zone heterotopias [172] cTKO ^1^: diffuse patterning of hippocampal layers [224]		deficits in glia endfoot formation/spanning [172] ↓axonal fibers [172,224] ↓axonal/dendritic outgrowth + branching [224]		disrupted in marginal zone heterotopias [172] cTKO ^1^: disrupted in hippocampus but not cortex [224]		disrupted synaptophysin staining in marginal zone heterotopias [172] cTKO ^1^: ↓spine density [224]
**Mena^–/–^**	normal [72] Mena ^–/–^ profilin^+/–^:prenatal lethal [72]		n. d.		↓axonal fibers [72,225] Mena ^–/–^ profilin ^+/–^: defects in neurulation [72]		n. d.		n. d.
**Mena^–/–^ VASP^–/–^** **EVL^–/–^**	perinatally lethal [76]		lissencephaly type II [75,76] exencephaly [76] altered cortical intralayer neuron positioning [76]		deficits in glia endfoot formation [76] ↓axonal fibers [75,76] ↓axonal/dendritic outgrowth [75,76]		disrupted in marginal zone heterotopia [75,76]		n. d.
**Genotype**		**NMJ Formation**		**Morphological Abnormalities**		**Behavior and Learning**		**Electrophysiology**	
**p97Fe65^–/–^ (↑p60Fe65** [15])		n. d.		normal [15]		impaired learning/memory [15,166]		↓LTP [166]	
**Fe65^–/–^**		↓pre-/postsynaptic area [169] ↓apposition of pre-/postsynapse [169] ↑fragmentation of postsynapse [169]		normal [167,169]		subtle muscle weakness [168,169] impaired learning/memory [169]		↓PTP [169], ←LTP [169], ←PPF [169]	
**Fe65L1^–/–^**		↓pre-/postsynaptic area [169] ↓apposition of pre-/postsynapse [169] ↑fragmentation of postsynapse [169]		normal [167,169] except: preliminary stage of cataract [168]		subtle muscle weakness [168,169] impaired learning/memory [169]		←PTP [169], ←LTP [169]	
**Fe65^–/–^/Fe65L1^–/–^**		↓pre-/postsynaptic area [169] ↓apposition of pre-/postsynapse [169] ↑fragmentation of postsynapse [169] centralized nuclei in muscle fibers [168]		↓body size [167] lens degeneration [168,169]/cataract [168] ↑ventricle size [167] ↓fimbria size and medial shift [167]		bilateral circling behavior [167,169] muscle weakness [168,169] impaired learning/memory [169] altered social behavior [169]		↓PTP [169], ↓LTP [169], ←PPF [169]	
**APP^–/–^** (↑APLP1 [205], ↑APLP2 [205])		Normal [226]		↓body weight [207,208,227,228] ↓brain weight [213,228] delayed eye opening [208] hypersensitivity to kainate-induced seizures [229]		muscle weakness [206,207,208,214,228] impaired learning/memory [207,209,214,218,223,227,228] altered social/innate behavior [208,209,228]		↓PTP [216], ↓LTP [214,216,217,228] ←↓PPF [208,227,230,231], ←mEPSC frequency [218], ←mEPSC amplitude [218], ↓mIPSC frequency [218], ←mIPSC amplitude [218]	
**APP^–/–^ APLP1^–/–^APLP2^–/–^**		n. d.		cTKO ^1^: ←cortical [224,232]/ hippocampal [232] volume		cTKO ^1^: impaired learning/memory [224,232] altered social/innate behavior [224]		cTKO^1^: ↓LTP [224,232], ↑↓PPF [224,232], ←↑mEPSC frequency [224,232], ←↑mEPSC amplitude [224,232], ←mIPSC frequency [224], ←↑mIPSC amplitude [224]	
**Mena^–/–^**		n. d.		n. d.		n. d.		n. d.	
**Mena^–/–^VASP^–/–^** **EVL^–/–^**		n. d.		exhibit edema [233] enlarged ventricle [76]		n. d.		n. d.	

^1^ Conditional triple knockout (TKO) of excitatory forebrain neurons; n. d., not determined; APP, amyloid precursor protein; APLP, APP-like protein; Fe65L, Fe65-like protein; Mena, mammalian enabled; VASP, vasodilator-stimulated phosphoprotein; EVL, Ena-VASP-like protein; CR, Cajal Retzius; CSPGs, chondroitin sulfate proteoglycans; MEF, mouse embryonic fibroblast; LTP, long term potentiation; PTP, post-tetanic potentiation; PPF, paired-pulse facilitation; mEPSC, miniature excitatory postsynaptic current; mIPSC, miniature inhibitory postsynaptic current; ↓, reduction; ↑, increase; ←, no alteration.

## Data Availability

Not applicable.

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
