# Peer review of "Fe65: A Scaffolding Protein of Actin Regulators"

_cells, 2021, doi:10.3390/cells10071599_

Round 1

Reviewer 1 Report

Prof Kins is an expert in FE65 and APP biology. In this review, Augustin and Kins summarize the interesting findings from different research groups concerning the putative role of FE65 in actin dynamics which has not been reviewed systematically elsewhere.  The review is written in a concise manner with a logically built concept and emphasizes several actin-related FE65-interactors and how FE65 might orchestrate their functions.

I suggest including a table to show the FE65 interactors that participate in actin dynamics.

FE65 is suggested to be a bridging molecule between surface receptors and the actin cytoskeleton. Therefore, ApoE receptors and NMDA receptors might be mentioned in Figure 1 too.

Fe65 is a phosphoprotein and some of its functions are regulated by phosphorylation. Previous publication has shown that Fe65 is a substrate of ERK1/2 (C.L. Standen, at el., 2003), which are important kinases for regulating actin polymerization (M.C. Mendoza, at el., 2015; S. Tanimura & K. Takeda, 2017). The author may consider discussing the potential of Fe65 phosphorylation in regulating the actin cytoskeleton remodeling.

Author Response

Point 1: I suggest including a table to show the FE65 interactors that participate in actin dynamics.

We included a corresponding table as “Table 1. Fe65 Interaction Partners Involved in Actin Dynamics” in our manuscript (page 4 line 118).

Point 2: Fe65 is suggested to be a bridging molecule between surface receptors and the actin cytoskeleton. Therefore, ApoE receptors and NMDA receptors might be mentioned in Figure 1 too.

We included an additional transmembrane receptor in Figure 1, representative for other putative receptors like the ApoE receptor and mentioned it in the figure legend (page 12 line 404).

Point 3: Fe65 is a phosphoprotein and some of its functions are regulated by phosphorylation. Previous publication has shown that Fe65 is a substrate of ERK1/2 (C.L. Standen, at el., 2003), which are important kinases for regulating actin polymerization (M.C. Mendoza, at el., 2015; S. Tanimura & K. Takeda, 2017). The author may consider discussing the potential of Fe65 phosphorylation in regulating the actin cytoskeleton remodeling.

Thank you for pointing this out. We agree with this suggestion and we incorporated a text section regarding phosphorylation states of Fe65 under the heading “4. Scaffolding Protein as Actin Regulators” (page 7 line 269). We discussed different phosphorylation sites of Fe65 and their potential role for interactions and localization. Further, we included the recommended publications in our manuscript.

Reviewer 2 Report

In this review, Augustin and Kins described the role of Fe65 protein family in the regulation of actin cytoskeleton dynamics. The review is well written and clear. My suggestions to improve the quality of the review are:

- to indicate that sAPPalpha has a neuroprotective effect, but not sAPPbeta in the Box1; the authors should therefore describe APP processing

- cite APP role in the synaptogenesis process, that can involve also actin dynamics, and describe AICD transgenic mice

- line 215 please define the abbreviation AMPAR

Author Response

Point 1: To indicate that sAPPalpha has a neuroprotective effect, but not sAPPbeta in the Box1; the authors should therefore describe APP processing.

We included a short text section about APP processing, going shortly into the different functions of sAPPalpha and sAPPbeta (page 2 line 67).

Point 2: cite APP role in the synaptogenesis process, that can involve also actin dynamics, and describe AICD transgenic mice

In our review, we describe in detail the overlapping phenotypes of Fe65 and APP family KO mice, which are presumably related to actin polymerization function including the APP phenotypes in synapse formation (chapter 5 and Table 2  page 9/column: “Synapse formation”). Furthermore, we point out the relevance of actin polymerization in the context of synaptic plasticity at various places in our manuscript (for example page 3, line 103; page 5, line 179; page 6 line 218; page 11, line 379; page 11, line 396). However, we have not discussed the different APP transgenic mice that overexpress different isoforms and/or mutant APP variants under different spatial and temporal conditions. Mainly, because the models do not show a coherent picture of APP or AICD function, possibly explained by differences in expression levels and promoter choice. Although we agree that this is an interesting topic and important for a better understanding of APP function, it is clearly beyond the scope of our review of Fe65 function in actin regulation. However, we have now included a paragraph pointing to the studies using Fe65 transgenic mice, including studies using combined co-expression with AICD, that highlight a role for the intracellular APP domain in actin regulation (page 7 line 288).

Point 3: line 215 please define the abbreviation AMPAR

We introduced the abbreviation for AMPAR in our manuscript (page 6 line 222).

Reviewer 3 Report

In this review paper, the authors first focused on Fe65 and its family proteins and summarized the similarity and difference among them. Then, focusing on binding proteins, they summarized the function of Fe65 related to gene transcription, which is still controversial. Next, they discussed the function of Fe65 as an actin regulator by focusing on the binding proteins as well. Importantly, they well-summarized the function-related observations, from cell to mouse, with deep insight. Since Fe65 is an adaptor protein and has many binding proteins, it may have a wide variety of physiological functions. Therefore, I think it is very useful to focus on single aspect of its functions and deeply discuss for it. This review satisfied this point of view.

I have some minor comments as below.

In page 3 line 91 to 92, after the sentence "the topic is still controversially discussed", please  cite some references including EMBO Rep. 2006 Jul;7(7):739-45. doi: 10.1038/sj.embor.7400704.

Please remove "So," from sentences in page 3 line 94, page 3 line 134, page 5 line 212, page 6 line 264 and page 8 line 292, 

The letters in Figure 1 are very small and difficult to see them. Although this is dependent on the editor's decision, I think that they should be larger to be recognized.

Author Response

Point 1: In page 3 line 91 to 92, after the sentence "the topic is still controversially discussed", please cite some references including EMBO Rep. 2006 Jul;7(7):739-45. doi: 10.1038/sj.embor.7400704

As suggested by the referee, we added the recommended as well as two additional references (page 3, line 93).

Point 2: Please remove "So," from sentences in page 3 line 94, page 3 line 134, page 5 line 212, page 6 line 264 and page 8 line 292

We removed “So,” and substituted it with other conjunctions (page 3 line 95, page 4 line 139, page 6 219, page 7 line 298, page 8 line 323).

Point 3: The letters in Figure 1 are very small and difficult to see them. Although this is dependent on the editor's decision, I think that they should be larger to be recognized.

We magnified the letters in Figure 1 and we hope that this have improved the readability (page 12 line 401).